# High sensitivity variable-temperature infrared nanoscopy of conducting oxide interfaces

Weiwei Luo [1,2], Margherita Boselli[1,2], Jean-Marie Poumirol[1], Ivan Ardizzone[1], Jérémie Teyssier[1], Dirk van der Marel[1], Stefano Gariglio [1], Jean-Marc Triscone[1] & Alexey B. Kuzmenko [1]

Probing the local transport properties of two-dimensional electron systems (2DES) confined at buried interfaces requires a non-invasive technique with a high spatial resolution operating in a broad temperature range. In this paper, we investigate the scattering-type scanning near field optical microscopy as a tool for studying the conducting $LaAlO_3/SrTiO_3$ interface from room temperature down to 6 K. We show that the near-field optical signal, in particular its phase component, is highly sensitive to the transport properties of the electron system present at the interface. Our modeling reveals that such sensitivity originates from the interaction of the AFM tip with coupled plasmon–phonon modes with a small penetration depth. The model allows us to quantitatively correlate changes in the optical signal with the variation of the 2DES transport properties induced by cooling and by electrostatic gating. To probe the spatial resolution of the technique, we image conducting nano-channels written in insulating heterostructures with a voltage-biased tip of an atomic force microscope.

[1] Department of Quantum Matter Physics, University of Geneva, Quai Ernest-Ansermet 24, 1211 Geneva, Switzerland. [2]These authors contributed equally: Weiwei Luo and Margherita Boselli. Correspondence and requests for materials should be addressed to A.B.K. (email: Alexey.Kuzmenko@unige.ch)

Two-dimensional electron systems (2DES) at oxide interfaces[1–3] offer a unique playground combining the physics of strong electronic correlations, typical of complex oxides, with quantum confinement induced by the finite thickness of the 2DES. This particular setting, being different from the one of semiconductor heterostructures regarding the symmetry of the electronic states, the carrier effective masses and other physical quantities, has been predicted to display novel phenomena. The experimental observation of a negative electronic compressibility for LaAlO₃/SrTiO₃ (LAO/STO) interfaces[4] as well as the evolution of the superconducting critical temperature upon gate tuning[5–7] have been interpreted as manifestations of such unique entanglement. This environment can also promote the coexistence of multiple electronic phases[8]. For instance, it was predicted that, in the presence of a Rashba-type spin–orbit interaction, whose relevance for the LAO/STO interface has been revealed in magnetotransport[9,10] and spin-charge conversion experiments[11,12], the 2D correlated electron system undergoes an electronic phase separation with an intrinsically inhomogeneous charge distribution[13]. Visualizing such effects remains challenging: it requires a local probe with nanometer-scale resolution capable of imaging the dynamic electric response of the 2DES. In this regard, optical probes are particularly interesting tools to investigate the electron dynamics[14,15]. In the far-field approach they, however, lack the spatial resolution, limited by the wavelength, which is the order of microns to millimeters in the infrared range used to detect the Drude response of the conduction electrons.

The rapidly developing technique of scattering-type scanning near-field optical microscopy (s-SNOM)[16,17] overcomes such limitation and allows the investigation of the amplitude and phase of the optical response with the resolution of 10–20 nm. This approach rests on the near-field interaction of the sample surface with a metal-coated atomic force microscope (AFM) tip illuminated by an infrared laser: the back-scattered light, recorded as a function of the tip position, brings information about the surface conductivity. Used to study local optical properties[18–21] and plasmon modes in metallic nanostructures[22–24] and two-dimensional materials[25–30], this approach has been also applied to oxide materials[31,32]. In particular, Cheng et al.[31] found that the presence of the 2DES at the LAO/STO interfaces can be detected using a room-temperature s-SNOM, opening new horizons for non-invasive near-field optical studies of the oxide interfaces. However, only the near-field amplitude was presented, without information about the phase component, and the authors argued that the increase of the near-field amplitude is a signature of a higher metallicity. Moreover, no systematic wavelength dependence was shown. As we show here, the 2DES-related amplitude change can be either positive or negative, depending on the wavelength and temperature.

Recently, a possibility to do s-SNOM measurements at low temperature has been demonstrated[33–35], with a great potential for studying complex phenomena in strongly correlated electron systems. Here we employ a cryogenic s-SNOM system (cryo-SNOM, neaspec GmbH) to study the near-field response of LAO/STO heterostructures in the range of wavelengths from 9.3 to 10.7 μm (CO₂ laser) from room temperature to 6 K. We demonstrate that both the amplitude and the phase of the back-scattered optical signal are sensitive to the presence of the 2DES at the LAO/STO interface. We consider a physical model of the near-field response arising from the formation of coupled plasmon–phonon modes at the interface. This model describes the wavelength dependence of the near-field signal and explains its temperature and field-effect evolution due to the variation of the charge mobility and density. Finally, we demonstrate that 160–190 nm wide conducting channels electrically created by an AFM tip can be detected in an insulating background, illustrating the high spatial resolution and high sensitivity of this technique applied to oxide interfaces.

## Results

### s-SNOM imaging of LAO/STO interface at room temperature.

LAO/STO heterostructures are prepared by pulsed laser deposition as described in the 'Methods' section. In order to identify the near-field signature of the 2DES, an insulating reference region is created on the crystalline STO (c-STO) substrate by depositing, using photolithography, an amorphous STO (a-STO) layer of 5–15 nm prior to the LAO growth. During the s-SNOM measurements the AFM is operated in tapping mode with an oscillation amplitude of about 60 nm and the detected signal is demodulated at a higher harmonic $n$ of the tapping frequency in order to suppress the far-field contribution to the back-scattering[17]. The tip is grounded to reduce the possible electro-static interaction with the sample. Interferometric detection allows measuring of both amplitude $s_n$ and phase $\phi_n$ of the near-field signal $s_n \exp(i\phi_n)$ (in this work we use $n = 3$). Figure 1 presents s-SNOM measurements at room temperature on two samples with a LAO layer thickness of 2 and 8 unit cells (u.c.), which are, respectively, below and above the threshold value (4 u.c.) for the formation of the conducting interface[2]. Notably, the laser wavelength used in the experiment, $\lambda_0 = 10.7$ μm (935 cm⁻¹), corresponds to a photon energy above the optical phonons in STO and LAO.

In the first sample (Fig. 1a), there is essentially no difference between the near-field signal (both in amplitude and phase) measured in the two insulating regions. In the second sample (b), the amplitude and phase contrasts between the conducting and insulating (reference) regions are remarkably strong: $s_3/s_{3,\mathrm{ref}} = 131\%$ and $\phi_3 - \phi_{3,\mathrm{ref}} = 0.36$ rad, where $s_{3,\mathrm{ref}}$ and $\phi_{3,\mathrm{ref}}$ are the reference values. The striking difference between the near-field images of the two samples proves that the contrast is due to the presence of the conducting 2DES. We note that not only the amplitude[31] but also the phase is a sensitive probe of the conducting electrons. These quantities are measured independently from each other and therefore bring complementary information.

### Theoretical model.

In order to understand the physical origin of the strong sensitivity of s-SNOM signal to the presence of the 2DES, we calculate the dispersion of the surface phonon–plasmon polaritons in the LAO/STO interface with and without 2DES for the in-plane momenta $q$ relevant for this experiment. In simulations, we use the optical dielectric functions of insulating LAO and STO extracted from previous far-field measurements[36,37] (Supplementary Note 1). They can be parameterized by a series of Lorentzians corresponding to optically active phonon modes (Supplementray Tables 1 and 2). Furthermore, considering the weak near-field contrast between the amorphous and crystalline STO (Supplementary Note 2), for simplicity we ignore a possible small difference between the dielectric functions of a-STO and c-STO in the infrared range. We model the dielectric function of the 2DES as the sum of insulating STO and of a Drude component[14,32]:

$$\varepsilon_{\mathrm{2DES}} = \varepsilon_{\mathrm{STO}} - \frac{n_{\mathrm{3D}}e^2}{m^*\varepsilon_0(\omega^2 + i\omega\tau^{-1})}, \quad (1)$$

where $\omega$ is the photon frequency, $m^*$ is the effective mass, $\varepsilon_0$ is the vacuum permittivity, $n_{\mathrm{3D}}$ is the three dimensional carrier density, $\tau^{-1} = \frac{e}{m^*\mu}$ is the scattering rate ($e$ is the elementary charge and $\mu$ is the optical carrier mobility). Based on the results of ab initio calculations[38] and previous infrared measurements[14], we use an exponential density distribution in

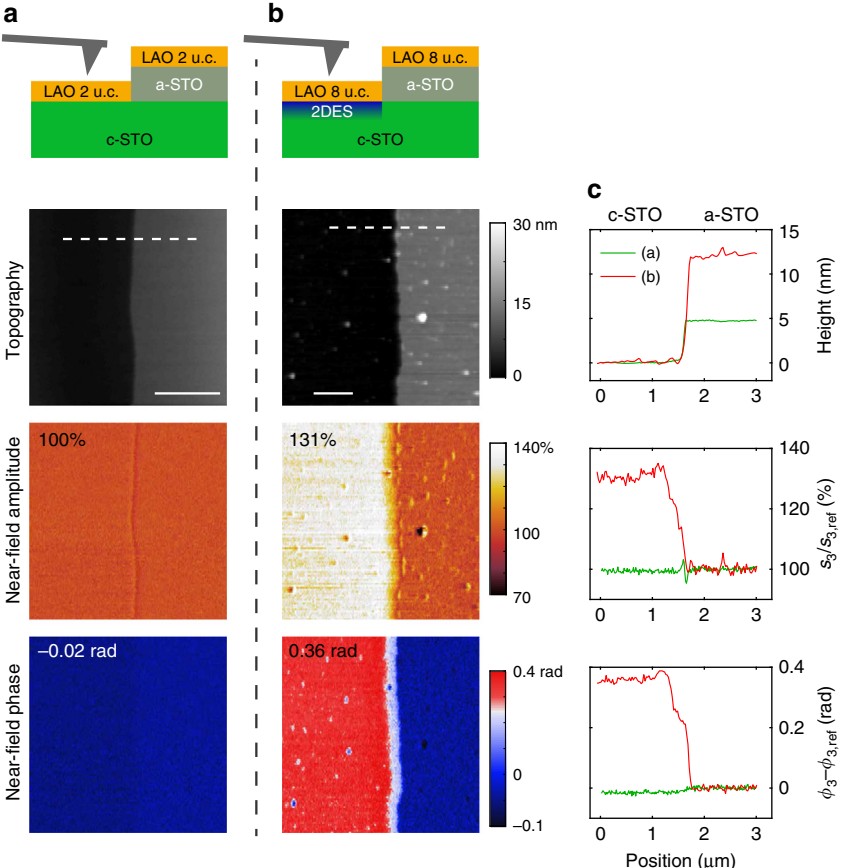

**Fig. 1** Demonstration of the sensitivity of s-SNOM to the presence of the 2DES at room temperature. **a, b** Sample description and images of the AFM height, near-field amplitude and near-field phase for the two samples with different thickness of LAO (2 and 8 u.c. respectively). The conducting 2DES is present only in (**b**), as the LAO thickness exceeds 4 u.c. **c** Line profiles along the dashed lines in (**a**) and (**b**). The near-field signal $s_3\exp(i\phi_3)$ at the third harmonic is normalized to the signal $s_{3,ref}\exp(i\phi_{3,ref})$ in the reference region with an additional layer of a-STO. The laser wavelength is 10.7 μm. The scale bars are 1 μm

the 2DES, $n_{3D}(z) = n_{2D}/z_0 \cdot \exp(-z/z_0)$, with the decay length $z_0 = 2$ nm, where the integrated 2D carrier density $n_{2D} = 8 \times 10^{13}$ cm$^{-2}$ is taken from transport experiments[39]. For the effective mass, we adopt the value $m^\star = 3.2 m_e$[14]. As it was noticed in previous infrared spectroscopy measurements[14,32], a much lower value for the mobility has to be used in the Drude model in doped strontium titanate to describe the data in the mid-infrared range, as compared with zero frequency transport experiments. This is related to the enhancement of the scattering rate due to electron–phonon interaction[14,40,41].

Figure 2a–c presents the imaginary part of the calculated reflection coefficient $r_p(q, \omega)$ (Supplementary Note 3) for a bulk STO covered with 2 nm of LAO without a 2DES (a) and hosting a 2DES (b, c) with two values of the optical carrier mobility, 10 cm$^2$ V$^{-1}$ s$^{-1}$ and 2 cm$^2$ V$^{-1}$ s$^{-1}$, respectively. In the first case, one can observe various surface phonon-polariton modes showing a weak dispersion due to a hybridization between the optical phonons in STO and LAO. The closest to our spectral range is the mode at 736 cm$^{-1}$ (marked as STO + LAO), which involves the highest-frequency optical phonons of both STO and LAO. The main effect of the presence of the 2DES (Fig. 2b) is the splitting of the STO + LAO mode into two branches. The lower one (labeled as LAO) does not disperse and originates from the LAO phonon. The upper one (called 2DES + STO) disperses strongly and is due to a plasmon mode in the 2DES, which is electromagnetically coupled to the phonon mode in STO. The interaction with the phonon sets the upper plasmon branch high

enough in energy to affect our measurement. Due to this coupling, the dispersion does not have the canonical $q^{1/2}$ expected for a 2D electron gas[42,43]. Interestingly, when the optical mobility is decreased to a rather small value of 2 cm$^2$ V$^{-1}$ s$^{-1}$ (Fig. 2c), the dispersion of the STO + 2DES branch becomes weaker, but the splitting remains visible. In Fig. 2d the real and imaginary parts of $r_p$ are shown at the optical momentum $q_{opt}$, where the tip-sample coupling function[26] (dashed curves in Fig. 2a–c) has a maximum. One can see that for both mobility values the presence of the plasmon mode influences significantly the reflection coefficient, especially its imaginary part, even though the mode frequency is below the optical range accessible with our laser (green region).

Based on the calculated $r_p(q, \omega)$, we simulate the s-SNOM spectra (Fig. 2e) for the three cases using a standard point-dipole model for the sample–tip interaction[26,44] (Supplementary Note 4). By comparison, one can see the influence of the 2DES on the near-field signal. Our calculations therefore explain the contrast observed in Fig. 1b as due to the appearance of a coupled plasmon–phonon mode. Interestingly, in the experimental spectral range the near-field phase in the presence of the 2DES is always above that of the non-conducting interface (the reference), showing a systematic increase with the mobility and decrease with the frequency. On the other hand, the amplitude shows a less systematic behavior.

**Temperature and frequency dependent s-SNOM measurements.** A direct way to test the theoretical prediction for the dependence of the optical response on the mobility is to measure

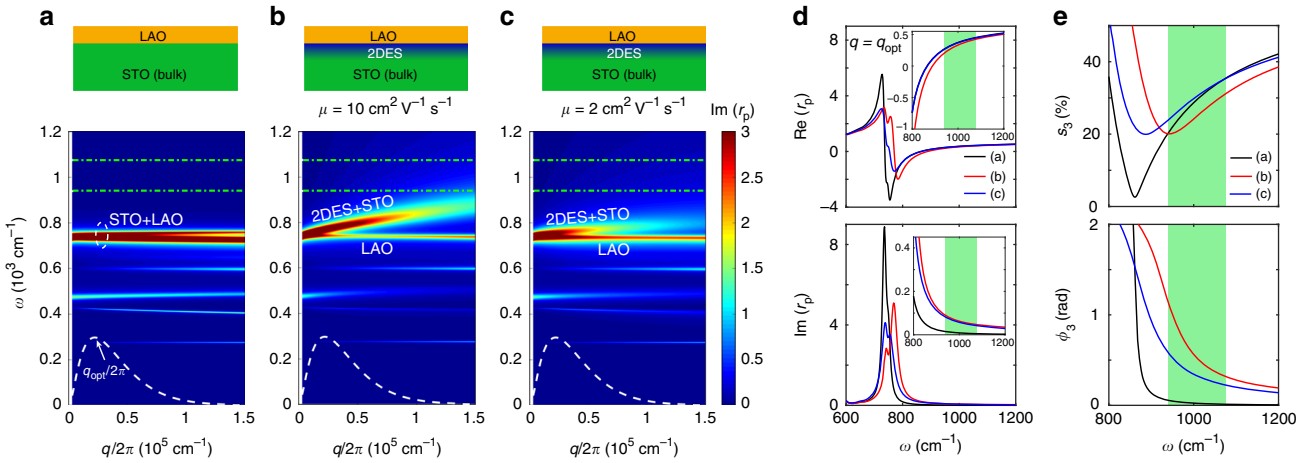

**Fig. 2** Theoretical dispersion of plasmon–phonon polaritons in the LAO/STO interface and the effect of the 2DES on the near-field response. **a–c** Optical dispersion for 2 nm of LAO on STO without a 2DES (**a**), LAO/STO containing a 2DES with the optical mobility of 10 cm$^2$ V$^{-1}$ s$^{-1}$ (**b**), and 2 cm$^2$ V$^{-1}$ s$^{-1}$ (**c**). The 2DES confinement is modeled by an exponential distribution of the 3D density with a total 2D carrier density of 8 × 10$^{13}$ cm$^{-2}$ and a confinement decay of 2 nm. The dashed white curves represent the momentum dependence of time-averaged near-field coupling weight function, which peaks at $q_{opt}/2\pi = 2.2 \times 10^4$ cm$^{-1}$. **d** The calculated real (top panel) and imaginary (bottom panel) parts of the reflection coefficient $r_p(\omega)$ at the optimal momentum, for the three cases a, b, and c. The insets show the frequency range around 1000 cm$^{-1}$. **e** Near-field amplitude (top panel) and phase (bottom panel) spectra normalized to a perfect metal (with $r_p = 1$) for the three cases calculated using the point-dipole model. The green dash-dotted lines in (**a–c**) and green regions in (**d, e**) show the spectral range in our experiment

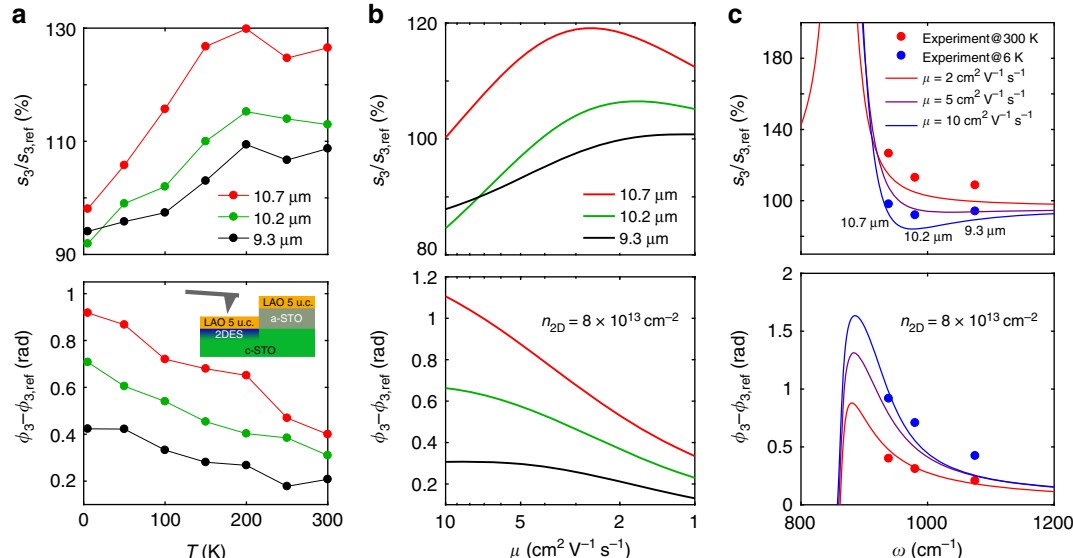

**Fig. 3** Temperature and frequency dependence of the near-field signal on the 2DES. **a** Experimental temperature dependence of the near-field amplitude and phase at laser wavelengths of 9.3, 10.2, and 10.7 μm. The inset in the bottom panel shows the schematic view of the sample with 5 u.c. of LAO. **b** Calculated dependence of the near-field signal on the optical mobility for the same wavelengths as in (**a**). **c** Calculated frequency dependence of the near-field signals for carrier mobility of 2, 5, and 10 cm$^2$ V$^{-1}$ s$^{-1}$. The experimental data at room temperature and 6 K are added for comparison. The 2D carrier density in (**b, c**) is 8 × 10$^{13}$ cm$^{-2}$

the temperature evolution of the near-field signal of the 2DES. Electrical DC transport measurements[1,39] reveal a dramatic increase of the carrier mobility and only a small decrease of the carrier density upon cooling. Figure 3a displays the temperature evolution of the amplitude and phase measured at three wavelengths, 9.3 μm (1075 cm$^{-1}$), 10.2 μm (980 cm$^{-1}$), and 10.7 μm (935 cm$^{-1}$). The corresponding near-field images at 6, 100, and 250 K are shown in Supplementary Note 5. In agreement with the calculations, the phase contrast is larger for the lowest frequency of light for any temperature. As the sample is cooled down, the near-field phase increases, reaching almost 1 rad at 6 K for the

10.7 μm wavelength. At the same time, the near-field amplitude decreases.

To compare the temperature dependence of the s-SNOM signals with the model, we compute the near-field response as a function of the optical mobility μ, while keeping all other parameters constant. In particular, in the frequency range of our experiment, the dielectric functions of STO and LAO do not change significantly with temperature[36,37]. In Fig. 3b, we show the calculated amplitude and phase contrasts for the three experimental wavelengths as a function of the mobility logarithmically decreasing from 10 to 1 cm$^2$ V$^{-1}$ s$^{-1}$. Such a

scale is used to accommodate the large increase of the DC mobility upon cooling[1,39] while taking into account the mentioned decrease of the mobility in the infrared range. A comparison with the experimental results confirms that the model correctly captures the thermal evolution of the amplitude and the phase signals at all studied frequencies. Notably the experimental values of the phase contrast are reproduced quantitatively by the calculations. The frequency dependence of the near-field amplitude and phase contrasts is shown in Fig. 3c for three selected values of $\mu$ (2, 5, and 10 cm$^2$ V$^{-1}$ s$^{-1}$). By plotting on the same graph the measured values at 300 K and 6 K for the three laser wavelengths we find a good agreement between the model and the experiment. This allows us to estimate that the optical mobility increases from ~2 cm$^2$ V$^{-1}$ s$^{-1}$ at room temperature to ~10 cm$^2$ V$^{-1}$ s$^{-1}$ at 6 K. In Supplementary Note 6 we show that the curves in Fig. 3b are robust with respect to a reasonable variation of the carrier density, the decay depth $z_0$ and the AFM parameters.

**Electrical tuning of the near-field response**. At low temperatures, electrostatic back-gating through the STO allows modulating the electrical properties of the 2DES[9,45]. A field-effect device based on a LAO/STO heterostructure with 5 u.c. of LAO is sketched in Fig. 4a. The DC sheet resistance (Fig. 4b, top) continuously increases as $V_G$ is tuned from zero toward negative values. This occurs due to a simultaneous decrease of the carrier density and mobility, as we determine from a separate Hall-effect measurement (Fig. 4b, bottom). Concomitantly the near-field response, measured at $\lambda_0 = 10.7\,\mu$m (Fig. 4c, d), shows a loss of contrast between the conducting and the reference regions. One can see that while the amplitude changes by only 5%, the phase decreases by more than a factor of 2 when $V_G$ is swept from 0 to −60 V.

The strong modulation of the phase and the weak variation of the amplitude with gating can be understood within the model described above. In Fig. 4e, we present a simulation where we use the experimentally determined gate-voltage dependence of the

carrier density (Fig. 4b) and the optical mobility by dividing the DC mobility by a factor of 20 in order to obtain reasonable LT optical mobility at $V_G = 0$. One can see that both trends are qualitatively reproduced. The small change of the amplitude is due to the opposite effect of the mobility and density on $s_3$ (see Supplementary Fig. 5). On the other hand, the decrease of both carrier density and mobility suppresses the near-field phase. This naturally explains the profound decrease of $\phi_3$ with the gate and corroborates our previous conclusion that the near-field contrast originates from the 2DES.

**Near-field imaging of AFM-written conducting wires**. Having demonstrated the sensitivity of the optical near-field technique to the electrical properties of the conducting layer, we investigate the capability of this approach to detect conducting regions at the nanoscale. To this purpose, we define nano-conducting channels in insulating heterostructures consisting of 3 u.c. of LAO on STO by applying a voltage bias to an AFM tip[46,47]. The conducting lines have widths ranging from tens to hundreds of nanometers, depending on the tip bias. The heterostructures used in these experiments are grown as described in the 'Methods' section and the writing canvas is prepared as described in previous works[48]. The wires are written using a MFP-3D Infinity AFM (Asylum Research-Oxford Instruments) in contact mode with a tip bias of 9 V and a scanning speed of 300 nm/s. Figure 5a shows s-SNOM images of two parallel conducting channels at room temperature. The conducting wires are clearly seen both in the amplitude and the phase images while the topographic AFM images (shown in Supplementary Fig. 6) do not reveal any structures at the wire positions. Albeit the absolute contrast is weak, a perpendicular line cut (Fig. 5b) nicely reveals the profile of the wires. Fitting the peaks with Lorentzians yields the full width at half maximum (FWHM) of 160–190 nm, which is reasonably close to an estimate (~120 nm) using the cutting method on a similar sample (Supplementary Note 7).

According to our simulation (Fig. 5c), one expects an experimentally detectable near-field contrast for carrier densities

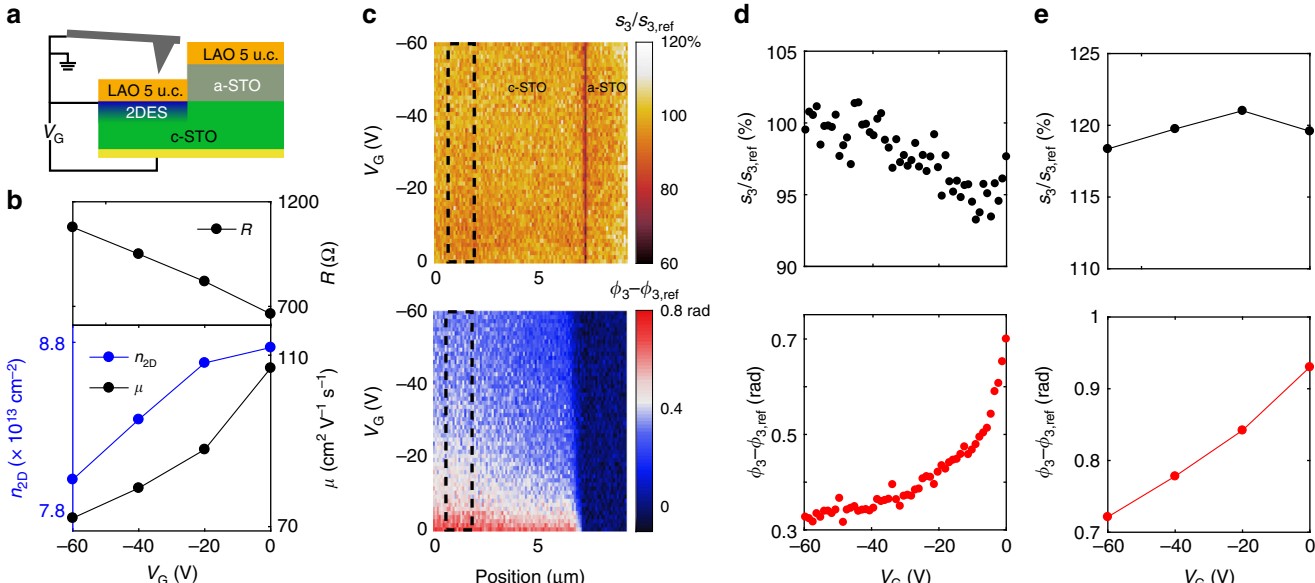

**Fig. 4** The effect of electrostatic gating on the electrical transport and near-field signal at 6 K. **a** Schematic view of the sample with a back gate. **b** The DC resistance, carrier density and mobility as a function of the gate voltage measured in an independent Hall-effect experiment on the same sample. **c** Experimental near-field amplitude and phase with respect to the reference region as a function of the position and the gate voltage. **d** The averaged near-field signals in the area marked by the dashed rectangles shown in **c** as a function of the gate voltage. **e** Calculation of the gate dependence of the near-field signal performed using the carrier density values and carrier mobility 20 times smaller than the DC values shown in (**b**). The wavelength is 10.7 μm

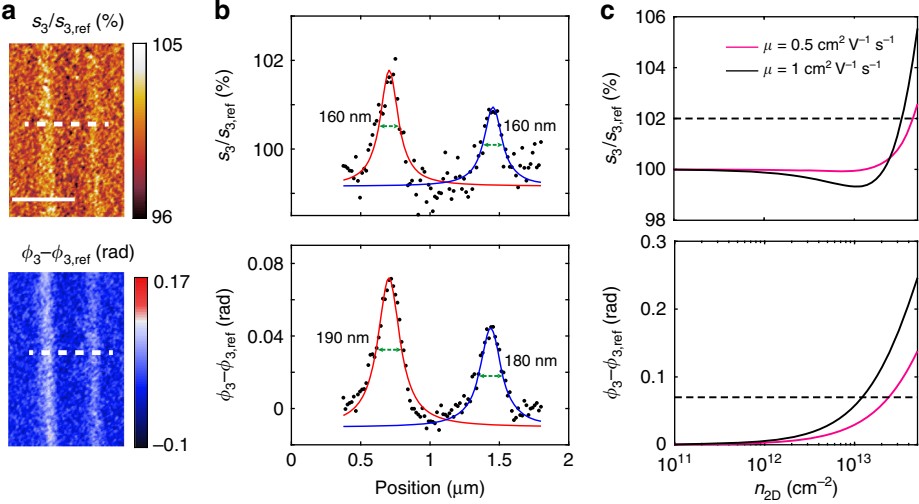

**Fig. 5** s-SNOM imaging of the AFM-written conducting wires in the LAO/STO interface with 3 u.c. of LAO. **a** Images of the near-field amplitude and phase, with respect to the signal from the regions not affected by the writing procedure. **b** The line profiles along the dashed lines in (**a**). **c** Calculated near-field signals on LAO/STO/2DES normalized by that on LAO/STO, as a function of the carrier density for optical mobility of 0.5 and 1 cm$^2$ V$^{-1}$ s$^{-1}$. The dashed lines indicate the near-field amplitude and phase values of the left wire in (**a**). Near-field measurements are performed at room temperature, about 80 min after the AFM-writing. The laser wavelength is 10.7 μm. The scale bar is 1 μm

above $10^{12}$ cm$^{-2}$. By comparing the measurements with the calculations, we find that the optical signal of the wires is well reproduced considering a carrier density of ~1 × 10$^{13}$ cm$^{-2}$ and a carrier mobility ranging from 0.5 to 1 cm$^2$ V$^{-1}$ s$^{-1}$, which is close to the optical mobility values found at room temperature in Fig. 3. The lower value of the carrier density in the written wires as compared with regular conducting oxide interface[39] is reasonable, taking into account that the s-SNOM measurements are performed about 80 min after the writing and that the conductivity of these devices at room temperature decreases in time[49]. In the future, one can envision the use of cryo-SNOM to freeze the written patterns immediately and conserve them long enough to perform a complete characterization.

## Discussion

The high sensitivity of the near-field signal to the parameters of the conducting interface is at first unexpected, since the Drude conductivity in the 2DES induces only a tiny change to the far-field infrared properties, such as the reflectivity and ellipsometric angles[14,15,50]. To qualitatively explain this striking difference, one should consider that the penetration depth of the far-field infrared radiation is of the order of tens to hundreds of microns, while the evanescent surface waves excited by the AFM tip in the s-SNOM experiment decay inside the material at distances comparable to the 2DES thickness. Therefore, whereas the far-field optical properties are dominated by the insulating strontium titanate, the near-field response is strongly influenced by the interface conductivity. The theoretical model developed here explains this effect quantitatively and moreover predicts the correct frequency, temperature and gate-voltage dependence of the near-field amplitude and the phase. In particular, we notice a rather systematic variation of the phase with respect to these parameters, making it a useful observable for mapping the 2DES. Moreover, we demonstrate that a higher amplitude does not necessarily signal a higher sample metallicity as suggested in ref. [31]. We believe that the use of more realistic treatments for the sample–tip interaction, such as the finite-dipole model[51,52] and extending the range of wavelengths will allow in the future the direct extraction of the 2DES parameters from the SNOM data.

Our study reveals the central role of the plasmon–phonon coupling in the oxide interfaces. While coupled surface phonon–plasmon modes were widely studied in conventional semiconductors[53], in the oxide interfaces they have been not explored so far, with an exception of the $q \sim 0$ Berreman mode observed in the far-field infrared spectra[14,15,50]. A related physical phenomenon is a clear observation of a huge decrease of the optical mobility above the phonon frequency as compared with the DC value, even though the absolute value of the optical mobility may be somewhat affected by the limited precision of the point-dipole model. Importantly, this observation is made here entirely on the basis of near-field measurements. This means that the electron–phonon interaction[37,40,41] is as efficient for the scattering of the surface phonon–plasmon modes as for the polaronic absorption at zero momentum.

While a high sensitivity of the near-field signal is important, the key advantage of s-SNOM over conventional optical techniques is a nanoscale spatial resolution. In the present case s-SNOM offers nanoscale imaging of the local metallicity linked to the carrier concentration and optical mobility. Here we exploit it for contactless near-field imaging of AFM-written conducting nanowires in the LAO/STO interface embedded in an insulating background. The ability to visualize buried nanoscale conducting structures shows clearly the usefulness of this technique for the development of oxide interface based electronics. It complements other non-invasive techniques such as piezoresponse force microscopy (PFM)[54], microwave impedance microscopy (MIM)[55], scanning SQUID microscopy[56], and scanning single-electron transistor (SET) microscopy[57] by offering nanoscale information about infrared optical response. We foresee that the use of this local optical probe, in combination with cryogenic performance and electrostatic gating, will provide important information on the possible phase separation and charge inhomogeneities due to ferroelectric domain walls[56–58], metal-insulator transitions and other emergent phenomena in a large family of 2D oxide interfaces[59,60].

## Methods

**Sample preparation.** The LAO/STO heterostructures were grown by pulsed laser deposition (PLD) on TiO$_2$-terminated (001) oriented commercial STO substrates (CrysTec GmbH). The samples were prepared in two steps: first a pattern was

defined with the photoresist on the substrates by photolithography and a layer of amorphous STO (a-STO) was deposited at room temperature in an atmosphere of $10^{-4}$ mbar of $O_2$ using a laser fluence of 0.6 J cm$^{-2}$ and a repetition rate of 1 Hz. After the removal of the photoresist that protected selected regions of the bare substrate from the deposition of a-STO, we grew the LAO films using the same experimental conditions but keeping the substrate at 800 °C. After the growth, the samples were annealed in situ for 1 h at 550 °C in an oxygen pressure of 200 mbar and cooled down to room temperature in the same atmosphere. The thickness of the LAO layer was monitored by reflection high energy electron diffraction (RHEED). The backside of one sample was covered with gold in order to realize a field-effect device.

## Data availability

The data that support the findings of this study are available from the corresponding author upon reasonable request.

## Code availability

All relevant calculation codes are available from the corresponding author upon reasonable request.

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

## Acknowledgements

This research was supported by the Swiss National Science Foundation (SNSF) through Division II and European Research Council (ERC) under the European Union's Seventh Framework Program (FP7/2007-2013)/ERC Grant Agreement no. 319286 (Q-MAC). The installation of the cryo-SNOM was supported by the Swiss National Science Foundation, the Rectorate and the Department of Quantum Matter Physics (DQMP) of the University of Geneva, the State Secretariat for Education, Research and Innovation (SERI), the Academic Society of Geneva, the Ernst and Lucie Schmidheiny Foundation, and the Ernest Boninchi Foundation. The authors thank J.L.M. van Mechelen for useful discussions and sharing his thesis data.

## Author contributions

W.L., M.B., D.vdM., and A.B.K. planned the experiments. W.L., M.B., and J.-M.P. carried out measurements. M.B., S.G., and J.-M.T. provided samples. W.L., I.A., J.T., and A.B.K. performed simulations. All authors discussed the data and contributed to writing the paper.

## Additional information

**Competing interests:** The authors declare no competing interests.

