## [Peer Review File · Nature Communications]

Reviewers' comments:

Reviewer #1 (Remarks to the Author):

- The present submission is an impressive technical demonstration of the capacity for cryogenic near-field optical microscopy to probe the electronic properties of "buried" oxide interfaces. With a clever sample preparation, the authors convincingly demonstrate bona fide optical sensitivity to the 2DEG manifesting at STO/LAO interfaces when the crystalline LAO layer exceeds a sufficient layer thickness. The predicted phenomenology of the 2DEG optical response follows from a sound analysis of the STO phonon-assisted plasma response. Systematic evolution of the STO/LAO heterostructure optical response under changing temperature, probing wavelength, and gate voltage establish impressive benchmarks for the sensitivity of near-field probes to the buried 2DEG. The authors demonstrate high spatial resolution to the optical response of conducting wires "written" electrostatically into the oxide interface, setting the stage for future application of this technique to resolve spatially inhomogeneous electronic properties emerging in the phenomenology of these oxide heterostructures. The work is of high quality and, at least with regards to the particular choice of application, so far unprecedented in the literature of near-field optical microscopy. I can recommend publication of this work in Nature Communications, provided first that the authors 1) resolve several minor omissions and modifications to the submission which I highlight in the following, and 2) adequately address a major caveat which I mention in closing:

- 1) Calls for minor modification:
- The authors provide adequate references for the general application of near-field optical microscopy as a probe of varied systems. However, the authors should resolve glaring omission of references to arguably the first convincing demonstrations of this method applied to correlated electron materials at low temperatures; in particular the authors should reference i) Yang, H. U. et al. "A cryogenic scattering-type scanning near-field optical microscope. Rev. Sci. Instrum. 84, 023701 (2013)." and ii) McLeod, A. S. et al. "Nanotextured phase coexistence in the correlated insulator V2O3." Nat. Phys. 13, 80–86 (2017), and perhaps also iii) the influential instrumental developments reported by the group of Lukas Eng at the Helmholtz-Zentrum Desden-Rossendorf.
- When simulating the optical response of the STO/LAO 2DES, the authors utilize a representative Drude optical permittivity with parameters carefully varying depth-wise across a 2 nm scale. Although this choice is well motivated, it may appear over-complicated for the present application, when the capacity of the near-field technique to resolve a conductance distribution over a 2 nm scale is already placed in question by relative insensitivity to the value z_0 and related parameters (as the authors discuss in supplement). Can the authors comment why this arguably complex model offers any advantage for interpreting nano-optical data compared with (for example) an effective sheet-conductance parametrization? The sheet conductance model is well established to rationalize the optical response of graphene, even when encapsulated ("buried") in optically transparent layers, similar to the present case of the 2DES. (See for instance Fei, Z. et al. Nano Lett. 11, 4701–4705 (2011), as well as Woessner, A. et al. Nat. Mater. 14, 421–425 (2015).).
- It is well reported that the so-called "point dipole" model affords a coarse description of the probe-sample near-field interaction; nevertheless the authors leverage this model for a qualitative interpretation of their data. Indeed, model predictions in Fig. 3a&b offer semi-quantitative agreement with the experimental findings. Meanwhile, the authors' spectroscopic explorations in Fig. 3c on the other hand show shortcomings of the simple point dipole model, a pathology considered and addressed by more realistic models as in e.g. Cvitkovic et al. Opt. Express 16, 8550–8565 (2007). The piecemeal agreement with experimental data is likely best mitigated by the authors' careful choice of free parameters in the point dipole model as mentioned in "Supplementary Note 4", including for instance the tip radius and the "distance of closest approach," which for quantitative considerations may be cause for concern. It would be instructive if the authors could demonstrate whether a more realistic treatment of the probe-sample near-field interaction (e.g. Hauer et al. Optics Expr. 20, 12 (2012)) might improve the spectroscopic agreement while (ideally) leaving unchanged the existing semi-quantitative agreements (including the "extracted" 5-fold improvement in mobility with decreased temperature). Although the authors' present treatment already demonstrates qualitative consistency with known properties of

the STO/LAO 2DES, a more realistic treatment of the optics problem will be imperative for future quantitative application of this technique to identify optical properties of embedded oxide heterostructures outside the extant well-studied regime. The authors should attempt a more realistic simulation of data in Fig. 3c, or comment why this remains conclusively beyond the scope of the present work.

- Can the authors justify more clearly why the temperature-dependent 2DES optical contrast of Fig. 3a matches qualitatively the simulations in Fig. 3b when the latter are plotted against logarithmically varying mobility? Since the near-field experiment can evidently be used as a sensitive probe of 2DES carrier mobility, what are we then to conclude from the present data about the temperature dependence of physical scattering mechanisms in the STO/LAO 2DES? The authors comment earlier "a much lower value for mobility has to be used in the Drude model in doped strontium titanate .. in the mid-infrared range"; it is not obvious whether this is a statement about behavior of the 2DES or simply about the STO substrate; the authors should clarify. Authors should also comment whether there is anything fundamentally new to be gleaned from monitoring the mid-IR mobility associated with these data, as compared with the well-reported temperature dependence of DC mobility of this system.

- The authors' explanation for unchanging amplitude of the 2DES optical response with concurrent evolution of the optical phase under applied gate hinges on notion of simultaneously changing mobility and carrier density. Can the authors advance or cite a model for this concurrent evolution that might offer qualitative agreement with the experimentally resolved optical contrasts in Fig 4d? At the very least, a schematic explaining the 2DES band structure and effective mass together with varying Fermi level would be most helpful for the reader to understand the physical basis for this reasoned phenomenology.

- Topographic AFM images are not shown for the sample surface after "writing" of conductive wires in the STO/LAO heterostructure, but it is not clear why this is so; these data should be presented in the supplementary material.

- Although the authors provide reference for the presumed temporal evolution of "written" conductive wires, this phenomenon merits further discussion in the manuscript to benefit the general audience. In fact, can the authors envision "writing" followed by immediate cooling of the sample within the same microscope apparatus to "freeze" these conducting regions in their pristine state? This would facilitate a more unambiguous characterization of the 2DES in heterostructures with thinner LAO layers.

- 2) Publication to Nature Communications calls for report of some impactful discovery either unreported or indicated ambiguously in previous work. This standard is slightly lacking in the presently submitted abstract, but with modification I believe the authors can meet this standard. I offer here one suggestion to resolve this "hole" in the present submission:

- Granted that the authors have demonstrated optical sensitivity to carrier density and mobility of the 2DEG, they go on to suggest the promise of this technique to resolve spatially varying optical conductivities of the 2DEG associated with inhomogeneous charge distributions, ferroelectric domain walls in STO, and other effects. Indeed, demonstrated imaging of conducting wires promises a spatial resolution at least as good as the wire width verified by the cutting method.

- The promise of high spatial resolution appears demonstrated again in the temperature- and wavelength-dependent near-field images presented in the supplemental section "Supplementary Note 5", which indicate a variation in the optical contrast from the 2DEG associated with atomic terraces in the STO substrate. This is a remarkable and under-presented finding, and should be highlighted in the main text. The authors should discuss this feature in the context of the optical conductivity analysis already leveraged for the "homogeneous" images considered almost exclusively therein. Does optical contrast at atomic terraces give suggestion for spatially modulating carrier density, carrier mobility, or both? Can the authors reason why the observed topographic steps could impart such electronic inhomogeneities? Possible influence of local strain at these terraces and/or uncompensated charge at the LAO-STO interface come to mind. Is it possible to quantify these inhomogeneities in terms of the operative Drude parameters?

- A warranted discussion observed inhomogeneities of the 2DES would substantiate the present work as a true demonstration for discovery science in oxide heterostructures.

Reviewer #2 (Remarks to the Author):

The authors present a method to non-invasively probe the conducting interface between two insulating complex oxides (LaAlO₃/SrTiO₃), with submicron resolution. Their data indicate an enhanced near-field signal when conducting electrons are present. They then present a model that indicates that their signal originates from plasmon-phonon coupling found in the conducting interface samples. Finally, they show conducting channels of ~150 nm width can be imaged by their non-invasive, near-field technique.

The paper is very well written. In the abstract and introduction, the authors write clearly about what they measure. In terms of originality, the authors themselves indicate that reference [31] uses an optical near-field probe to study a LAO/STO interface. My feeling is that the present manuscript does expand on that work in a meaningful way, but there should be more discussion of what was done in that reference and what is unique about the present work.

This work should be of interest to the complex oxide community, and perhaps may be of interest to a broad community, however it has some deficiency which preclude its publication in its present form. I outline some of my objections and suggestions below.

1. The experiments were carried out with ~10 μm CO₂ laser. There is no discussion of why this energy was chosen.
2. I find the argument on page 9-10 discussing why the amplitude contrast is a weak function of V_g and the phase a strong function of V_g not rigorous. I think the authors could remedy this in part by presenting Figure 4e in a different way (discussed more below).
3. There have been several other papers that discuss non-invasive imaging LAO/STO. PFM: Huang et al, APL Materials 1, 052110 (2013); MIM: Jiang et al, APL 111, 233104 (2017); Reference [49] can be discussed in the context of non-invasive probe (scanning SQUID); [49] also has a companion article by Honig that uses scanning SET. The authors should discuss how their near-field IR nanoscope is similar or different to the above.
4. Figures 1 is generally clear and compelling. However:
 - a. Text indicates that the amplitude contrast is given by $s_3/s_{3,\text{ref}} - 1$, while the figure and caption state it is $s_3/s_{3,\text{ref}}$
 - b. Is there independently measured transport data showing that the 8uc LAO film has a conducting interface?
5. Figure 4 becomes difficult to understand and raises some other questions about the research. A few comments:
 - a. What is the role of grounding the AFM tip as shown in panel (a)? How is the electrical connection to the tip important or not? Is the tip conducting? If it is important what was the configuration for the data presented in Figure 1 and 3?
 - b. Inset of panel (b) tries to show the device schematic but it is very difficult to distinguish the different parts of the Hall bar.
 - c. Why is amplitude contrast not as strong as shown in Figure 1? Similarly, why is the phase contrast stronger than shown in Figure 1?
 - d. Panel (d) shows the amplitude & phase vs V_g , and (e) shows model results vs carrier density and mobility, each of which is a function of V_g . I think it would be best to show how changing V_g moves to different positions on the 3 curves, or otherwise relate n & μ to V_g . Or show experimental data in (d) as function of n and μ ? Are the authors able to show experimentally how the near-field signals depend on n and μ ?
 - e. Why is the device shown in Figure 4 grown with 5 uc LAO rather than the 8 uc that was used for Figure 1?
6. Figure 5: The amplitude and phase contrast are apparent, but I think could be more obvious simply by changing the intensity scale. Supplementary Figure 5 shows some characterization of the AFM-written wire. Why is there a background conductance of 0.12 μS after the wire was cut? Could the background conductance contribute to the low contrast presented in Figure 5 vs 1?

7. In the conclusion the authors state that their technique will be useful for studying ferroelectric domain walls. But their measurement should already be sensitive to them (For example in Supplementary Figure 2). Did they see any evidence of such domain walls in the existing work? Why or why not?
8. The authors should relate the measured optical mobilities to what could be expected for a DC transport measurement.

Reviewer #3 (Remarks to the Author):

Luo et al.'s paper discusses the application of scattering-type scanning near-field optical microscopy (s-SNOM) to the study of the two-dimensional electron system (2DES) at the interface of LaAlO₃ and SrTiO₃, from room temperature down to 6K. 2D electron systems as a class of phenomena have not been extensively studied with s-SNOM, and the authors provide modeling for interpretation of the near-field optical signal. These are interesting experiments, and demonstrate a powerful and challenging temperature-dependent approach that extends on previous work studying 2DES at room temperature with s-SNOM (ref. 31 in the paper). However, there are several puzzling aspects of the paper that require more explanation and make it difficult to judge the impact of the work.

- The theoretical model is based on modeling the 2DES with an insulating STO and Drude response, incorporating dispersion due to a coupled plasmon-phonon polariton mode. The splitting of this mode is strongly dependent on the (optical) mobility of the carriers in the 2DES, so that the authors argue that for their excitation energy of $\sim 1000 \text{ cm}^{-1}$, the tail of the polariton mode will affect the near-field response. Based on the dispersion relations shown in Fig. 2a, b, c, this does not seem entirely convincing, and becomes less convincing for the calculations of the near-field response for a given dielectric function based on the point dipole model (Fig. d,e). For low mobilities, no difference is theoretically expected between the case of no 2DES and 2DES at this energy in the near-field amplitude, and only a small difference for the high mobility case. Furthermore, the near-field amplitude for a high-mobility 2DES appears to be lower than that for no 2DES. How does this then agree with the experimental Fig. 1b, with the 2DES leading to an increase in the near-field amplitude in comparison to the no 2DES case?

- Similarly, the authors claim the amplitude shows a less systematic behavior than the phase when measuring their 2DES materials. Do the authors have any explanation for why the amplitude is consistently higher than that predicted by the point dipole model?

- Fig. 1 shows near-field amplitude contrast between the LAO/c-STO (with an interfacial 2DES) and LAO/a-STO sections. Why do we not see similar amplitude contrast when the system is set at $V_g = 0$ for the gate voltage measurements? (Fig. 4c,d)

- Topographic artifacts are common in s-SNOM measurements, with cross-talk in the near-field due to fluctuations in the tip phase and amplitude frequently occurring when scanning over a topographic feature. The authors mention that they see no topographic features in their scans, but since the conducting wires mentioned on page 10 were written in contact mode at a relatively high voltage, there is a high probability they would introduce some topographic artifacts. The changes in the near-field shown in Figure 5 are relatively small, so ensuring that there is no topographic cross-talk is important.

- At the end of the paper, some additional work is included studying lithographically-formed nanochannels, to illustrate the spatial resolution of the technique. But overall, it is not clear to me whether the high spatial resolution provided by s-SNOM is valuable in understanding this class of materials. As the authors point out, "the optical response is spatially homogeneous away from the step" of the LAO-STO. The discussion emphasizes that the small penetration depth of the near-field is the key to its high sensitivity to the 2DES, which is a strong argument and should be mentioned earlier. However, the attribution of signal to coupled plasmon-phonon polariton modes implies that the broad wavevector distribution of the tip is also playing a vital role in the near-field contrast. Do the authors have evidence of this? For example, if the excitation frequency is above the (effective 3d) plasma frequency of the 2DES, this should result in high reflectivity without the

need to invoke polariton modes and tip-launching mechanisms. Is the near-field contrast arising from a fundamentally different mechanism than far-field studies?

Reviewer #1 (Remarks to the Author):

- The present submission is an impressive technical demonstration of the capacity for cryogenic near-field optical microscopy to probe the electronic properties of “buried” oxide interfaces. With a clever sample preparation, the authors convincingly demonstrate bona fide optical sensitivity to the 2DEG manifesting at STO/LAO interfaces when the crystalline LAO layer exceeds a sufficient layer thickness. The predicted phenomenology of the 2DEG optical response follows from a sound analysis of the STO phonon-assisted plasma response. Systematic evolution of the STO/LAO heterostructure optical response under changing temperature, probing wavelength, and gate voltage establish impressive benchmarks for the sensitivity of near-field probes to the buried 2DEG. The authors demonstrate high spatial resolution to the optical response of conducting wires “written” electrostatically into the oxide interface, setting the stage for future application of this technique to resolve spatially inhomogeneous electronic properties emerging in the phenomenology of these oxide heterostructures. The work is of high quality and, at least with regards to the particular choice of application, so far unprecedented in the literature of near-field optical microscopy. I can recommend publication of this work in Nature Communications, provided first that the authors 1) resolve several minor omissions and modifications to the submission which I highlight in the following, and 2) adequately address a major caveat which I mention in closing:

We are grateful for this highly favorable judgement of our work. All the recommendations of the Referee are taken into account in the revised version.

- 1) Calls for minor modification:

- The authors provide adequate references for the general application of near-field optical microscopy as a probe of varied systems. However, the authors should resolve glaring omission of references to arguably the first convincing demonstrations of this method applied to correlated electron materials at low temperatures; in particular the authors should reference i) Yang, H. U. et al. “A cryogenic scattering-type scanning near-field optical microscope. Rev. Sci. Instrum. 84, 023701 (2013).” and ii) McLeod, A. S. et al. “Nanotextured phase coexistence in the correlated insulator V2O3.” Nat. Phys. 13, 80–86 (2017), and perhaps also iii) the influential instrumental developments reported by the group of Lukas Eng at the Helmholtz-Zentrum Desden-Rosendorf.

Indeed! We added a sentence citing these three seminal works.

- When simulating the optical response of the STO/LAO 2DES, the authors utilize a representative Drude optical permittivity with parameters carefully varying depth-wise across a 2 nm scale. Although this choice is well motivated, it may appear over-complicated for the present application, when the capacity of the near-field technique to resolve a conductance distribution over a 2 nm scale is already placed in question by relative insensitivity to the value z_0 and related parameters (as the authors discuss in supplement). Can the authors comment why this arguably complex model offers any advantage for interpreting nano-optical data compared with (for example) an effective sheet-conductance parametrization? The sheet conductance model is well established to rationalize the optical response of graphene, even when encapsulated (“buried”) in optically transparent layers, similar to the present case of the 2DES. (See for instance Fei, Z. et al. *Nano Lett.* 11, 4701–4705 (2011), as well as Woessner, A. et al. *Nat. Mater.* 14, 421–425 (2015).).

This is a good question as indeed the sheet-conductance approximation works well in certain cases, such as graphene. However, we have several reasons to use a finite-thickness model. First, a (quasi-) exponential charge density distribution with the decay length of about 2 nm has been demonstrated experimentally (Dubroka et al., *PRL* 104, 156807 (2010)) and theoretically (Son et al., *PRB* 79, 245411 (2009)). Second, we base our simulation on the 3D dielectric function in the bulk STO, dominated by the phonons, which cannot be reduced to the 2D limit. Third, by forcing the conductivity to be two-dimensional we would *a priori* exclude relevant physical phenomena, such as the Berreman mode observed in the far-field studies by Dubroka et al.

Nevertheless, according to what the Referee implies, we cannot determine from our measurement the actual decay length, and we do not attempt to do this. To demonstrate the relative insensitivity of our present set of measurements to this parameter, in Fig.1 we compare a simulation, where $z_0 = 0.4$ nm is equal to one lattice constant, with the one used in the main text ($z_0 = 2$ nm). One can see that this simulation is also consistent with our data.

Fig. 1. The same calculation as in Fig. 3(b) for $Z_0 = 0.4$ nm (left) and $Z_0 = 2$ nm (right).

Fig. 2. The same calculation as in Fig. 3(b) ($Z_0 = 2$ nm), shown in linear (left) and logarithmic (right) mobility scale.

- It is well reported that the so-called “point dipole” model affords a coarse description of the probe-sample near-field interaction; nevertheless the authors leverage this model for a qualitative interpretation of their data. Indeed, model predictions in Fig. 3a&b offer semi-quantitative agreement with the experimental findings. Meanwhile, the authors’ spectroscopic explorations in Fig. 3c on the other hand show shortcomings of the simple point dipole model, a pathology considered and addressed by more realistic models as in e.g. Cvitkovic et al. *Opt. Express* 16, 8550–8565 (2007). The piecemeal agreement with experimental data is likely best mitigated by the authors’ careful choice of free parameters in the point dipole model as mentioned in “Supplementary Note 4”, including for instance the tip radius and the “distance of closest approach,” which for quantitative considerations may be cause for concern. It would be instructive if the authors could demonstrate whether a more realistic treatment of the probe-sample near-field interaction (e.g. Hauer et al. *Optics Expr.* 20, 12 (2012)) might improve the spectroscopic agreement while (ideally) leaving unchanged the existing semi-quantitative agreements (including the “extracted” 5-fold improvement in mobility with decreased temperature). Although the authors’ present treatment already demonstrates qualitative consistency with known properties of the STO/LAO 2DES, a more realistic treatment of the optics problem will be imperative for future quantitative application of this technique to identify optical properties of embedded oxide heterostructures outside the extant well-studied regime. The authors should attempt a more realistic simulation of data in Fig. 3c, or comment why this remains conclusively beyond the scope of the present work.

We are well aware that the extended dipole model addresses some limitations of the point dipole model. In the present case we have chosen to use the latter for several reasons. First, it was shown by Amarie and Keilmann (*PRB* 83, 045504 (2011)) that the point dipole model works relatively well in

the non-resonant regime away from phonon modes. Second, the extended dipole model introduces extra 3 to 4 parameters and therefore is significantly more elaborated. Given that the point dipole model explains qualitatively our data, we decided not to overcomplicate the analysis, which would make the manuscript less readable. Third, the extended dipole was applied most of the times for semi-infinite samples, except in the paper by Hauer et al mentioned by the Referee. Therefore, we feel that using it in our case (multilayer sample) is too early.

At the same time, we fully agree with the Referee, that in the future the quantitative analysis of the SNOM data will require going beyond the point dipole model, especially in the range of phonon resonances. Therefore, we added a corresponding sentence to the Discussion, and cited the References mentioned by the Referee.

- Can the authors justify more clearly why the temperature-dependent 2DES optical contrast of Fig. 3a matches qualitatively the simulations in Fig. 3b when the latter are plotted against logarithmically varying mobility? Since the near-field experiment can evidently be used as a sensitive probe of 2DES carrier mobility, what are we then to conclude from the present data about the temperature dependence of physical scattering mechanisms in the STO/LAO 2DES? The authors comment earlier “a much lower value for mobility has to be used in the Drude model in doped strontium titanate in the mid-infrared range”; it is not obvious whether this is a statement about behavior of the 2DES or simply about the STO substrate; the authors should clarify. Authors should also comment whether there is anything fundamentally new to be gleaned from monitoring the mid-IR mobility associated with these data, as compared with the well-reported temperature dependence of DC mobility of this system.

Plotting the data in linear scale (Fig.2, left) does not change the conclusions, however we choose to plot the data in logarithmic scale (Fig.2, right) as the experimental trend is reproduced slightly better. Moreover, in the literature (former Ref. [48] and [14]), the DC mobility is usually shown in a log scale as a function of T, because it changes by several orders of magnitude between 300 K and 6 K.

The statement “a much lower value for mobility has to be used in the Drude model in doped strontium titanate in the mid-infrared range” is an experimental observation previously seen both in the 2DES (Dubroka et al., PRL 104, 156807 (2010)) and doped bulk STO (Lewin et al., Adv. Funct. Mater. **28**, 1802834 (2018)).

At this stage, we limit ourselves by crosschecking the first cryo-SNOM results with the existing information about the temperature dependence of the physical parameters of the 2DES. We sincerely hope that with a broader spectral range and a more realistic modelling we will be able to extract the temperature dependence of the 2DES parameters directly from the SNOM data.

- The authors' explanation for unchanging amplitude of the 2DES optical response with concurrent evolution of the optical phase under applied gate hinges on notion of simultaneously changing mobility and carrier density. Can the authors advance or cite a model for this concurrent evolution that might offer qualitative agreement with the experimentally resolved optical contrasts in Fig 4d? At the very least, a schematic explaining the 2DES band structure and effective mass together with varying Fermi level would be most helpful for the reader to understand the physical basis for this reasoned phenomenology.

We significantly improved this explanation in the text. Most importantly, in Fig. 4b, we added the gate dependence of the carrier density and DC mobility measured on the same sample in an independent Hall-effect experiment, which clearly shows that both quantities decrease when gate voltage is swept towards negative values. At the same time, we decided to replace in Fig.4b the two-terminal resistance measured during the SNOM experiment with the 4-terminal resistance in the mentioned Hall setup. Furthermore, we moved Fig.4e to the Supplementary Information and added there a simulation based on the experimentally measured carrier density and the DC mobility, which we scaled down to approximately match the values used by us to explain the temperature dependent data. The comparison between Fig. 4d and 4e is now much more convincing.

- Topographic AFM images are not shown for the sample surface after “writing” of conductive wires in the STO/LAO heterostructure, but it is not clear why this is so; these data should be presented in the supplementary material.

Following the Referee’ suggestion, we now present these images in the Supplementary information.

- Although the authors provide reference for the presumed temporal evolution of “written” conductive wires, this phenomenon merits further discussion in the manuscript to benefit the general audience. In fact, can the authors envision “writing” followed by immediate cooling of the sample within the same microscope apparatus to “freeze” these conducting regions in their pristine state? This would facilitate a more unambiguous characterization of the 2DES in heterostructures with thinner LAO layers.

This is an excellent idea. Unfortunately, for technical reason we cannot, at present, write the wires in the SNOM setup and we do it externally (as is mentioned in the text). Therefore, we cannot start freezing the wires immediately after writing them. We thank the Referee for this suggestion and we add a sentence to the main text: “In the future, one can envision using cryo-SNOM for freezing the written patterns and thus conserving them for a longer time needed for a complete characterization.”

- 2) Publication to Nature Communications calls for report of some impactful discovery either unreported or indicated ambiguously in previous work. This standard is slightly lacking in the presently submitted abstract, but with modification I believe the authors can meet this standard. I offer here one suggestion to resolve this “hole” in the present submission:

- Granted that the authors have demonstrated optical sensitivity to carrier density and mobility of the 2DEG, they go on to suggest the promise of this technique to resolve spatially varying optical conductivities of the 2DEG associated with inhomogeneous charge distributions, ferroelectric domain walls in STO, and other effects. Indeed, demonstrated imaging of conducting wires promises a spatial resolution at least as good as the wire width verified by the cutting method.

We mention the potential of this technique for studying domain walls in the discussion (see also our answer to the second Referee).

- The promise of high spatial resolution appears demonstrated again in the temperature- and wavelength-dependent near-field images presented in the supplemental section “Supplementary Note 5”, which indicate a variation in the optical contrast from the 2DEG associated with atomic terraces in the STO substrate. This is a remarkable and under-presented finding, and should be highlighted in the main text. The authors should discuss this feature in the context of the optical conductivity analysis already leveraged for the “homogeneous” images considered almost exclusively

therein. Does optical contrast at atomic terraces give suggestion for spatially modulating carrier density, carrier mobility, or both? Can the authors reason why the observed topographic steps could impart such electronic inhomogeneities? Possible influence of local strain at these terraces and/or uncompensated charge at the LAO-STO interface come to mind. Is it possible to quantify these inhomogeneities in terms of the operative Drude parameters?

We thank the Referee for outlining the importance of this effect. Indeed, understanding the terrace effect is important and interesting. However, we prefer to keep these images in the Supplementary Information because at this moment we do not have enough experimental data to make definitive physics claims on the effect of terraces on the near-field spectra. We are working on collecting more experimental information on this issue and the related modelling, which we plan to publish in a separate paper.

- A warranted discussion observed inhomogeneities of the 2DES would substantiate the present work as a true demonstration for discovery science in oxide heterostructures.

Indeed, we extensively discuss the SNOM imaging of controlled inhomogeneities, such as AFM written wires.

Reviewer #2 (Remarks to the Author):

The authors present a method to non-invasively probe the conducting interface between two insulating complex oxides (LaAlO₃/SrTiO₃), with submicron resolution. Their data indicate an enhanced near-field signal when conducting electrons are present. They then present a model that indicates that their signal originates from plasmon-phonon coupling found in the conducting interface samples. Finally, they show conducting channels of ~150 nm width can be imaged by their non-invasive, near-field technique.

The paper is very well written. In the abstract and introduction, the authors write clearly about what they measure.

Thanks a lot.

In terms of originality, the authors themselves indicate that reference [31] uses an optical near-field probe to study a LAO/STO interface. My feeling is that the present manuscript does expand on that work in a meaningful way, but there should be more discussion of what was done in that reference and what is unique about the present work.

In the revised version we list in the introduction the numerous developments presented in our work as compared to ref.[31]. Moreover, we correct some non-rigorous statements made in ref.[31]

This work should be of interest to the complex oxide community, and perhaps may be of interest to a broad community, however it has some deficiency which preclude its publication in its present form. I outline some of my objections and suggestions below.

1. The experiments were carried out with ~10 μm CO₂ laser. There is no discussion of why this energy was chosen.

Unfortunately, we can only use the wavelengths between 9.3 and 10.7 microns, as we have only a CO₂ laser as a light source. In the future it would be nice to extend the spectral range, which is a costly development.

2. I find the argument on page 9-10 discussing why the amplitude contrast is a weak function of V_g and the phase a strong function of V_g not rigorous. I think the authors could remedy this in part by presenting Figure 4e in a different way (discussed more below).

The remark is in line with a comment of Referee 1. Following the advices of both Referees, we changed the Fig. 4, added more data and modified the related discussion. Now our original arguments are justified much better.

3. There have been several other papers that discuss non-invasive imaging LAO/STO. PFM: Huang et al, APL Materials 1, 052110 (2013); MIM: Jiang et al, APL 111, 233104 (2017); Reference [49] can be discussed in the context of non-invasive probe (scanning SQUID); [49] also has a companion article by Honig that uses scanning SET. The authors should discuss how their near-field IR nanoscope is similar or different to the above.

We thank the Referee for pointing out these relevant papers. We now mention them in the Discussion and emphasize the complementarity of s-SNOM with respect to other non-invasive techniques.

4. Figures 1 is generally clear and compelling. However:

a. Text indicates that the amplitude contrast is given by $s_3/s_{3,ref} - 1$, while the figure and caption state it is $s_3/s_{3,ref}$

this is fixed

b. Is there independently measured transport data showing that the 8 μ c LAO film has a conducting interface?

We characterized all our samples using transport measurements. To address this question, we now present in the Supplementary Figure 3 the curve $R(T)$ for the sample used in temperature-dependent SNOM measurement.

5. Figure 4 becomes difficult to understand and raises some other questions about the research.

As it is indicated above, we significantly modified this figure to make it more understandable.

A few comments:

a. What is the role of grounding the AFM tip as shown in panel (a)? How is the electrical connection to the tip important or not? Is the tip conducting? If it is important what was the configuration for the data presented in Figure 1 and 3?

The tip is grounded in all our measurements in order to reduce the possible electrostatic interaction with the sample that would affect the AFM performance. The tip is conducting as it is covered by metal as required in the s-SNOM technique. In the revised version we added a sentence about grounding.

b. Inset of panel (b) tries to show the device schematic but it is very difficult to distinguish the different parts of the Hall bar.

We agree and we remove this picture as it is not important for the paper.

c. Why is amplitude contrast not as strong as shown in Figure 1? Similarly, why is the phase contrast stronger than shown in Figure 1?

These two measurements are done at different temperatures. The difference is consistent with the T-dependence presented in Fig. 4a.

d. Panel (d) shows the amplitude & phase vs V_g , and (e) shows model results vs carrier density and mobility, each of which is a function of V_g . I think it would be best to show how changing V_g moves to different positions on the 3 curves, or otherwise relate n & μ to V_g . Or show experimental data in (d) as function of n and μ ? Are the authors able to show experimentally how the near-field signals depend on n and μ ?

In the new version of this figure we present the experimentally measured carrier density and DC mobility as a function of the gate voltage using Hall effect. Now panel (d) shows a simulation of the near-field data in panel (c) using this information. Now the comparison is straightforward.

e. Why is the device shown in Figure 4 grown with 5 uc LAO rather than the 8 uc that was used for Figure 1?

For technical reasons, we used different samples for Figure 1 and Figure 4. They show rather similar metallic DC resistivity and near-field response.

6. Figure 5: The amplitude and phase contrast are apparent, but I think could be more obvious simply by changing the intensity scale.

We changed the intensity scale, and indeed it looks better.

Supplementary Figure 5 shows some characterization of the AFM-written wire. Why is there a background conductance of 0.12 μS after the wire was cut? Could the background conductance contribute to the low contrast presented in Figure 5 vs 1?

The residual conductance after the cut is explained by the choice of the tip bias during the cut. However, the Referee raises a very important point as we cannot exclude that the contrast between the wire and the background might be affected by the properties of the heterostructures used for the AFM-writing experiments. These samples have a 3 u.c. thick LAO layer, therefore they are about to develop the 2DES and the optical response of the wire background might be slightly different from that stemming from an insulating heterostructure (for instance STO covered by an amorphous film).

We added this information into the Supplementary file.

7. In the conclusion the authors state that their technique will be useful for studying ferroelectric domain walls. But their measurement should already be sensitive to them (For example in Supplementary Figure 2). Did they see any evidence of such domain walls in the existing work? Why or why not?

[redacted]

[redacted]

[redacted]

8. The authors should relate the measured optical mobilities to what could be expected for a DC transport measurement.

It is known from the literature (Dubroka et al., PRL 104, 156807 (2010), Lewin et al., Adv. Funct. Mater. **28**, 1802834 (2018)) that the optical mobility is much smaller than the DC mobility. We mention this in the text.

Reviewer #3 (Remarks to the Author):

Luo et al.'s paper discusses the application of scattering-type scanning near-field optical microscopy (s-SNOM) to the study of the two-dimensional electron system (2DES) at the interface of LaAlO₃ and SrTiO₃, from room temperature down to 6K. 2D electron systems as a class of phenomena have not been extensively studied with s-SNOM, and the authors provide modeling for interpretation of the near-field optical signal. These are interesting experiments, and demonstrate a powerful and challenging temperature-dependent approach that extends on previous work studying 2DES at room temperature with s-SNOM (ref. 31 in the paper).

We thank the Referee for this positive judgement.

However, there are several puzzling aspects of the paper that require more explanation and make it difficult to judge the impact of the work.

- The theoretical model is based on modeling the 2DES with an insulating STO and Drude response, incorporating dispersion due to a coupled plasmon-phonon polariton mode. The splitting of this mode is strongly dependent on the (optical) mobility of the carriers in the 2DES, so that the authors argue that for their excitation energy of $\sim 1000 \text{ cm}^{-1}$, the tail of the polariton mode will affect the near-field response. Based on the dispersion relations shown in Fig. 2a, b, c, this does not seem entirely convincing, and becomes less convincing for the calculations of the near-field response for a given dielectric function based on the point dipole model (Fig. d,e). For low mobilities, no difference is theoretically expected between the case of no 2DES and 2DES at this energy in the near-field amplitude, and only a small difference for the high mobility case.

Indeed, a small contrast is seen in Fig. 2d (which is the reflection coefficient at a fixed momentum). However, the point-dipole model result is shown in Fig. 2e, where a finite contrast is present.

Furthermore, the near-field amplitude for a high-mobility 2DES appears to be lower than that for no 2DES. How does this then agree with the experimental Fig. 1b, with the 2DES leading to an increase in the near-field amplitude in comparison to the no 2DES case?

The room temperature data shown in Fig.1b correspond better to the low-mobility case (2 cm²/Vs). For this mobility, the amplitude of 2DES is higher than the amplitude of no 2DES at the frequency used in Fig.1b (10.7 microns), which agrees qualitatively with the SNOM data. However, it is true that the experimental contrast (about 130%) is higher than theoretical prediction (115%). Such a difference might be due to limitations of the point-dipole model.

- Similarly, the authors claim the amplitude shows a less systematic behavior than the phase when measuring their 2DES materials. Do the authors have any explanation for why the amplitude is consistently higher than that predicted by the point dipole model?

As just mentioned, the point dipole model is only an approximation. As we reply to Referee 1, using more sophisticated treatments, such as finite-dipole model is premature at this stage. Nevertheless, we hope that in the future this will help obtaining a better quantitative agreement. This is now mentioned in the Discussion section.

- Fig. 1 shows near-field amplitude contrast between the LAO/c-STO (with an interfacial 2DES) and LAO/a-STO sections. Why do we not see similar amplitude contrast when the system is set at $V_g = 0$ for the gate voltage measurements? (Fig. 4c,d)

The reason is that Fig.1 shows data at room temperature while Fig.4 refers to a gate dependence at 6 K (since the efficiency of gating through STO at room temperature is strongly reduced).

- Topographic artifacts are common in s-SNOM measurements, with cross-talk in the near-field due to fluctuations in the tip phase and amplitude frequently occurring when scanning over a topographic feature. The authors mention that they see no topographic features in their scans, but since the conducting wires mentioned on page 10 were written in contact mode at a relatively high voltage, there is a high probability they would introduce some topographic artifacts. The changes in the near-field shown in Figure 5 are relatively small, so ensuring that there is no topographic cross-talk is important.

This is an excellent point, also raised by Referee 1. We added the topography data, as well as the tip amplitude and phase, into the Supplementary Information.

- At the end of the paper, some additional work is included studying lithographically-formed nanochannels, to illustrate the spatial resolution of the technique. But overall, it is not clear to me whether the high spatial resolution provided by s-SNOM is valuable in understanding this class of materials. As the authors point out, "the optical response is spatially homogeneous away from the step" of the LAO-STO.

In the supplementary information we briefly mention the inhomogeneity related to the atomic terrace steps. This clearly shows the high resolution of s-SNOM and its ability to probe even small sample inhomogeneities. However, the physical origin of this modulation is not well established, which does not allow us to discuss it in the main text. As a related remark, we showed to Referee 2 the ability of SNOM to image the domain walls (which deserves a separate publication).

The discussion emphasizes that the small penetration depth of the near-field is the key to its high sensitivity to the 2DES, which is a strong argument and should be mentioned earlier.

As the Referee suggests, we now mention this explicitly in the abstract.

However, the attribution of signal to coupled plasmon-phonon polariton modes implies that the broad wavevector distribution of the tip is also playing a vital role in the near-field contrast. Do the authors have evidence of this? For example, if the excitation frequency is above the (effective 3d) plasma frequency of the 2DES, this should result in high reflectivity without the need to invoke polariton modes and tip-launching mechanisms. Is the near-field contrast arising from a fundamentally different mechanism than far-field studies?

Indeed, the mechanisms of near-field and far-field optical response and contrast are fundamentally different. In the far field measurements, the electromagnetic waves with essentially zero in-plane k-vector propagate deep into the sample. In the s-SNOM, the tip excites a broad distribution of surface waves with different and large in-plane momenta. To demonstrate the Referee this difference, in Fig.4 we present simulated normal-incidence far-field reflectivity spectra of LAO/STO with and without 2DES. One can see that the contrast introduced by 2DES in the far field is tiny, in agreement with previous experiments (Dubroka et al., PRL 104, 156807 (2010). Nucara et al., PRB 97, 155126 (2018)).

Fig. 4. Simulated far-field reflectivity spectra of the LAO/STO sample without 2DES and with 2DES (mobility 10 and 2 cm²/Vs) as in the former Fig.2 of the main text.

List of changes (the page numbers correspond to the previous version):

Main text:

- Abstract: the sentence “Our modelling reveals ..” is modified to mention the small penetration depth
- Page 2: a few sentences are added to the Introduction clarifying the new developments of our work as compared to ref.[31].
- Page 2: the sentence “Recently, a possibility to do s-SNOM measurements at low temperature has been demonstrated, with a great potential for studying complex phenomena in strongly correlated electron systems.” is added.
- Page 3: the sentence is added: “The tip is grounded in order to reduce the possible electrostatic interaction with the sample.”
- Figures 4b, 4e and 5a are modified and the relevant text is corrected
- Page 11: the sentence “In the future, one can envision the use of cryo-SNOM to freeze the written patterns immediately and conserve them long enough to perform a complete characterization.” is added in the end of the section about the near-field imaging of conducting wires.
- Page 12: the sentence “The use of more realistic treatments for the sample tip-interaction, such as the finite-dipole model \cite{cvitkovic2007analytical,hauer2012quasi} and extending the range of wavelengths will hopefully allow in the future the direct extraction of the 2DES parameters from the SNOM data.” is added.
- Page 12: the sentence “It complements other non-invasive techniques such as piezoresponse force microscopy (PFM) \cite{huang2013direct}, microwave impedance microscopy (MIM) \cite{jiang2017direct}, scanning SQUID microscopy \cite{kalisky2013locally} and scanning single-electron transistor (SET) microscopy \cite{honig2013local} by offering nanoscale information about infrared optical response.” is added
-

Supplementary Information

- Figure 2: a panel with the curve $R(T)$ is added, with a corresponding text.
- Figure 5 and 6 are added, with a corresponding text.
- Page 9: two sentences: “A small tip bias (-3 V) applied to the tip during the cut improves the resolution of the technique, however it is not enough to pinch the conducting wire. As a consequence, the total conductance is reduced, but not completely suppressed by the cut.” are added.

REVIEWERS' COMMENTS:

Reviewer #1 (Remarks to the Author):

The authors have satisfactorily addressed nearly all the inadequacies identified in my my first review report Therefore I recommend publication to Nature Communications, provided first that the authors provide clarification pursuant to the following ambiguity that remains in their work:

For their simulations to render agreement with their experimental data, the authors employ an optical mobility an order of magnitude lower than the DC mobility identified for functionally identical samples. The authors provide tentative justification for this choice based on infrared measurements of the 2DES by Dubroka et al., PRL 104 (2010). However, in that work, optical mobilities of the 2DES for similar LAO/STO samples (with LAO thickness similar to that considered by the present authors) was scarcely more than a factor of two lower than corresponding DC mobilities. It is meanwhile true that lower optical mobilities were reported for samples with much thicker LAO top layer, but relevance of those results to samples in the present work is not at all clear.

In sum, since the low-temperature nano-imaging contrast demonstrated here is evidently an impressive signifier of mobility in such a 2DES, the authors should devote several more sentences to discuss the potential import of their "low optical mobility findings". Should nano-imaging uncover inhomogeneities in the local mobility, what physical properties of the 2DES might these be ascribed to? More critically, the authors should clarify whether readers are to take seriously the identified mobilities on any quantitative basis, or are these perhaps underestimated by the simulations due to inadequacies in the point dipole model? With a more adequate disambiguation of this matter, the present work can and should be published to Nature Communications.

Reviewer #2 (Remarks to the Author):

I thank the authors for their responses and for privately sharing the domain walls images. In the answer to Question 4b the authors refer to "Supplementary Figure 3" but I think they mean "2".

I am satisfied with the revisions to the manuscript and recommend publication.

Reviewer #3 (Remarks to the Author):

The authors have responded in detail to the reviewer comments and I am satisfied with their responses.

For their simulations to render agreement with their experimental data, the authors employ an optical mobility an order of magnitude lower than the DC mobility identified for functionally identical samples. The authors provide tentative justification for this choice based on infrared measurements of the 2DES by Dubroka et al., PRL 104 (2010). However, in that work, optical mobilities of the 2DES for similar LAO/STO samples (with LAO thickness similar to that considered by the present authors) was scarcely more than a factor of two lower than corresponding DC mobilities. It is meanwhile true that lower optical mobilities were reported for samples with much thicker LAO top layer, but relevance of those results to samples in the present work is not at all clear.

In fact, Dubroka et al introduced *two types* of the optical mobility. The first value has been extracted by fitting of the Drude peak in the far-infrared range (Fig.1 b in their paper). The second (mid-infrared) value has been obtained from the analysis of the Berreman mode between 800 and 1200 cm⁻¹ (Fig. 3c). The mid-infrared optical mobility appears to be much smaller than the far-infrared value as the authors mention explicitly:

Finally, we comment on the marked difference between $\mu^{\text{IR}} = 34 \text{ cm}^2/\text{V s}$ as derived from the Berreman mode at 900 cm^{-1} and the corresponding $\mu^{\text{IR}} = 700 \text{ cm}^2/\text{V s}$ from the Drude response at low frequency. Notably, a similar frequency dependence of the mobility, or a strong inelastic contribution to the scattering rate, was observed in bulk $\text{SrTi}_{1-x}\text{Nb}_x\text{O}_3$ [19] and explained in terms of polaronic correlations.

We believe that in this comment the Reviewer is referring to the far-infrared value from this paper. On the other hand, it would be more appropriate to compare our optical mobility with the *mid-infrared value* from Dubroka et al, since it was obtained on the data in the same spectral range, as ours. The mid-infrared mobility obtained by Dubroka et al is between 10 and 34 cm²/Vs at low temperature, depending on the model used. The first value matches almost quantitatively our result at 6 K (Fig.3). Therefore, we believe that our results are consistent with Dubroka et al.

In sum, since the low-temperature nano-imaging contrast demonstrated here is evidently an impressive signifier of mobility in such a 2DES, the authors should devote several more sentences to discuss the potential import of their "low optical mobility findings". Should nano-imaging uncover inhomogeneities in the local mobility, what physical properties of the 2DES might these be ascribed to? More critically, the authors should clarify whether readers are to take seriously the identified mobilities on any quantitative basis, or are these perhaps underestimated by the simulations due to inadequacies in the point dipole model? With a more adequate disambiguation of this matter, the present work can and should be published to Nature Communications.

We fully agree. In order to address this pertinent remark and to remove any ambiguity, we added four clarifying sentences to the Discussion:

"A related physical phenomenon is a clear observation of a huge decrease of the optical mobility above the phonon frequency as compared to the DC value, even though the absolute value of the optical mobility may be somewhat affected by the limited precision of the point-dipole model. Importantly, this observation is made here entirely on the basis of near-field measurements. This means that the electron-phonon interaction [37,40,41] is as efficient for the scattering of the surface phonon-plasmon modes as for the polaronic absorption at zero momentum. ... In the present case s-SNOM offers nanoscale imaging of the local metallicity linked to the carrier concentration and optical mobility."